# Factors Related to Health Risk Communication Outcomes among Migrant Workers in Thailand during COVID-19: A Case Study of Three Provinces

**DOI:** 10.3390/ijerph182111474

**Published:** 2021-10-31

**Authors:** Ratchadaporn Papwijitsil, Hathairat Kosiyaporn, Pigunkaew Sinam, Mathudara Phaiyarom, Sataporn Julchoo, Rapeepong Suphanchaimat

**Affiliations:** 1Field Epidemiology Training Program (FETP), Division of Epidemiology, Department of Disease Control, Ministry of Public Health, Nonthaburi 11000, Thailand; rapeepong@ihpp.thaigov.net; 2International Health Policy Program (IHPP), Ministry of Public Health, Nonthaburi 11000, Thailand; hathairat@ihpp.thaigov.net (H.K.); pigunkaew@ihpp.thaigov.net (P.S.); mathudara@ihpp.thaigov.net (M.P.); sataporn@ihpp.thaigov.net (S.J.)

**Keywords:** risk communication, migrants, practice, awareness, COVID-19, Thailand

## Abstract

Coronavirus disease 2019 (COVID-19) is a newly emerging infectious disease, and risk communication is one of several public health emergency responses. During the pandemic, many migrant workers in Thailand experienced barriers that hamper access to health information. This study aims to explore factors related to the outcomes of health risk communication, including awareness of public health measures and preventive practices. We conducted a cross-sectional survey on migrants between January and April 2021 using cluster sampling in Phuket, Ranong, and Samut Sakhon. In the descriptive analysis, we presented the median, proportion, and ratio, while in the inferential analysis, we employed a logistic regression with robust standard errors. Although a total of 303 participants were initially included in this study, the final number was narrowed down to 288 samples due to insufficient information required for the analysis. Frequent reception of health information and primary school education showed a statistically significant association with preventive practices. Middle-aged migrant workers demonstrated a significantly lower level of preventive practices than younger migrant workers. A longer stay in Thailand was significantly related to a lower degree of awareness toward public health measures. Thus, it is necessary to promote the accessibility of health information among migrant workers in Thailand, especially those who have lived in Thailand for more than eight years, are older, and have no formal education.

## 1. Introduction

The coronavirus disease 2019 (COVID-19) was first recognized in late 2019 [1,2,3] and was later declared by the World Health Organization (WHO) on 30 January 2020 to be a Public Health Emergency of International Concern (PHEIC) [4]. In response, the WHO aimed to launch a global coordinated effort for effective preparedness and response to COVID-19. According to a WHO report on 20 October 2021, the COVID-19 pandemic has affected over 241 million people and caused 4.9 million deaths in more than 200 countries [5]. 

Thailand was the first country outside China to report the presence of COVID-19. The first wave of COVID-19 in Thailand started with clusters of infections related to imported cases from other countries and local transmission [6]. In response to this, the Thai Government introduced several non-pharmaceutical interventions to curb the outbreak, for instance, international travel bans and social distancing [7]. Risk communication was included in the Incident Command System (ICS) as an essential component for health emergency preparedness and response [8]. The Thai Government established the Center for COVID-19 Situation Administration (CCSA) as the governing body of the ICS [9]. Due to these combined measures and a resilient health system, Thailand succeeded in containing the first wave of COVID-19 in 2020 [10,11].

However, from late December 2020, Thailand faced a new wave that was more severe than the previous one [12]. This second wave was believed to have originated from migrant workers in the inner city of Samut Sakhon [13], a province within the vicinity of Bangkok and a major residential area hosting a large number of migrants. Thailand is known as one of the popular cross-broader destinations in Southeast Asia, and the number of migrants in Thailand was approximately 4.9 million in 2018; most of the migrants relocated from neighboring countries, including Cambodia, Lao PDR, Myanmar, and Viet Nam (CLMV nations) [14], with the greatest number of migrants coming from Myanmar [15].

The vulnerability of migrants has been highlighted during the COVID-19 pandemic. Most migrants live in crowded housing with poor hygiene and have limited capacity for social distancing, leading to the spread of infection [16]. Migrants may also experience difficulties in accessing health information and services due to financial hardship and language barriers [17]. 

Prior studies in Thailand have found that migrant health workers (MHWs), migrant health volunteers (MHVs), and village health volunteers (VHVs) were utilized in the Government’s initiative to provide access to health information to migrants, especially for pandemic preparedness in previous emerging diseases such as influenza (H1N1) [18,19]. They also found that while most migrants had a high level of knowledge about the disease, they did not agree with preventive measures such as border control or social distancing. Moreover, education and reception of information from MHW and television were significantly associated with higher knowledge about health issues. During COVID-19, the International Organization for Migration conducted rapid studies in 2020 and found that approximately one-third of migrants did not understand health information [20,21,22]. Studies that explore factors relating to behavioral outcomes and health risk communication specifically among migrants in Thailand during COVID-19 are also lacking.

To address this gap in knowledge, it is necessary to investigate factors influencing the reception of health information. The health belief model (perceived risk, perceived severity, perceived benefits, and perceived barriers) was used as an overarching concept for this study [23]. Therefore, this study aims to explore the outcomes of health risk communication, including the level of awareness and self-reported COVID-19 prevention among migrants, along with other associated factors. It is hoped that the findings from this study will provide impactful contributions toward public health research on migrant health in Thailand and result in more comprehensive migrant health policies.

## 2. Materials and Methods

### 2.1. Study Design and Setting

A cross-sectional survey was conducted between January and April 2021 on migrant workers aged 15 years and above in the three provinces of Phuket, Ranong, and Samut Sakhon. These provinces were chosen since they have some of the most densely migrant-populated areas in Thailand. Surveys were distributed to survey participants in the main district of each province. 

### 2.2. Sample Size and Sampling Method

Cluster sampling was used as the basis for obtaining our samples. We purposively selected two main migrant communities in each province that were identified by local providers or representatives of non-government organizations (NGOs) who were familiar with the field. Then, we randomly selected households based on the proportion of households in the community compared with the total number of households in the study. We randomly selected a migrant from each household (likely to be either the household head or the household head’s spouse). The sample size calculation was based on the following formula, n = deff × (Z^2^*P*(1 −
*P*)/e^2^) where deff (design effect) = 2; Z for 5% type-1 error = 1.96; *P* = 0.96 (proportion of people with regular hand washing during COVID-19) [24], e = 0.05. According to this formula, the calculated sample size was 242 but we expanded the sample size to 303 to account for a 20% non-response rate or incomplete responses. 

### 2.3. Data Collection

A bilingual (Thai and Myanmar) interviewer-assisted questionnaire was used in this study. We asked MHWs and MHVs in the field to help with the interviews. The interviewers described the purpose, risks, and benefits of the study and obtained informed consent from participants before collecting data. The questionnaire was mainly paper-based, but online questionnaires were also provided to participants for their convenience. 

### 2.4. Questionnaire Design

The questionnaire consisted of three sections: (i) personal information; (ii) awareness toward public health measures on COVID-19 and self-reported practices during COVID-19; (iii) questions about health literacy levels, modes, and frequency to obtain health information, the health belief model, and other potential confounders such as knowledge about disease and attitude toward public health measures.

Personal information consisted of questions about gender, age, ethnicity, family status, occupation, education, income, length of stay in Thailand, medical insurance, Thai language fluency, and resources supporting access to information and practices. 

Questions about knowledge and self-reported practices for COVID-19 were adapted from the Department of Disease Control (DDC), Thailand’s website [25], while attitude and awareness of public health measures on COVID-19 were adapted from CCSA’s announcement [26]. Health literacy questions were adapted from Osborne et al. [27] and questions for assessing constructs of the health belief model were modified from the version found in Jones et al. [23].

The questionnaire was pilot tested and underwent content validation by five experts. Then, we calculated the index of item objective congruence (IOC) and revised the questionnaire until each question had an IOC score of more than 0.5. We conducted a 30-participant pilot survey in Phuket to test the questionnaire’s reliability and clarity. We calculated Cronbach’s alpha coefficient for each set of questions based on the theme and found that all of them exceeded 0.7, reflecting satisfactory reliability. An instructional manipulation check (IMC) question was placed in the middle of the questionnaire to ensure that the participants had adequately read the instructions for all questions.

### 2.5. Data Analysis

For descriptive analysis, we used median and percentiles for continuous data, and proportions and ratios were used for categorical data. In the inferential analysis, we used logistic regression with robust standard error for both univariate and multivariate analyses. STATA^®^ version 14 was used for the calculation. In the multivariate analysis, we selected only variables that had a *p*-value ≤ 0.1 from the univariate analysis into the model and presented the results in the form of an adjusted odds ratio (AOR) with a 95% confidence interval (CI).

The analytic part consisted of two main strands: (i) awareness of public health measures and (ii) self-reported preventive practices during COVID-19. For the first strand, demographic data and sources of information were designated as independent variables, whereas awareness of public health measures on COVID-19 served as a dependent variable. For the latter strand, the independent variables covered demographic data, resources for supporting preventive practices, the overall frequency of receiving health information from all sources, health literacy, knowledge about the disease, awareness, and attitudes toward public health measures, and the four constructs of the health belief model (perceived risk, perceived benefit, severity of illness, and perceived barriers), while self-reported preventive practices toward COVID-19 was considered a dependent variable (Table 1). The arrangement of each variable is presented in Table 2.

## 3. Results

### 3.1. Descriptive Analysis

A total of 303 participants were recruited, but 15 were excluded from the analysis due to failure to follow protocol such as not following the IMC question or not providing their age. Consequently, the final number of participants was 288; of those, 8 were from Phuket (3%), 76 were from Ranong (26%), and 204 were from Samut Sakhon (71%). The median age of the participants was 30 (25th percentile = 27, 75th percentile = 37), and the male-to-female ratio was 1:2.1. The median length of stay in Thailand was eight years. In terms of ethnicities, most of the participants were Myanmar (63%), followed by Mon (27%), Dawei (8%), and Karen (2%). Most of them had completed primary school (58%). Regarding occupation, factory workers (49%) constituted the majority of the participants. Most of the participants (65%) had income levels lower than the provincial minimum wage. In terms of Thai comprehension, 55% of the participants were partially fluent in speaking Thai, while 62% of the participants could not read Thai. For insurance, the majority of participants (64%) were insured by the Social Security Scheme (the main public insurance scheme for formal-sector workers). About one-fifth (22%) were uninsured, and approximately 13% were insured by the Health Insurance Card Scheme (the main public insurance scheme for informal-sector migrants, managed by the Ministry of Public Health). 

A large proportion of participants exhibited a high level of access to resources supporting exposure to health information. Approximately 99% of participants had access to electricity, 94% had phone signals, and 92% had internet connectivity. Access to resources supporting preventive practices was also high, with 99% of the participants having access to face masks and about 88% having access to alcohol-based sanitizers; access to soap and tap water was also high, at approximately 97% and 94%, respectively. In terms of information dissemination, social media was the most common source of health information (67%); the second and third most common sources were from MHW/MHV/VHV (63%) and health professionals (56%), respectively (see Figure 1). 

The majority of participants reported a high score on many questions such as awareness of public health measures (85%) and preventive practices (77%), followed by knowledge (68%), attitude (63%), and health literacy (58%). The percentage of participants that reported receiving health information frequently from all sources combined was 54%. For the health belief model constructs, most participants had low scores for perceived susceptibility (75%), perceived barriers (71%), and perceived severity (66%). More details are shown in Figure 2.

### 3.2. Inferential Analysis

Univariate analysis found a significant variation in the awareness of public health measures based on the length of stay in the country (*p*-value = 0.03). Other independent variables for awareness of public health measures did not show statistically significant associations (see Table 3).

For preventive practices, gender (*p* = 0.04), age groups (*p* = 0.03), education (*p* = 0.05), occupation (*p* = 0.03), Thai reading comprehension (*p* < 0.01), health insurance (*p* < 0.01), frequency of receiving health information (*p* < 0.01), and perceived barriers (*p* = 0.03) showed statistically significant associations. More details are described in Table 4.

We included the length of stay in Thailand and health insurance status in the multivariate analysis, as both variables had a *p*-value of less than or equal to 0.10 in the univariate analysis. The multivariate analysis revealed that a long length of stay in Thailand (more than 8 years (median)) was significantly associated with low awareness of public health measures to tackle the COVID-19 situation (AOR = 0.43, 95% CI 0.19–0.95) (Table 5). 

For the effects on preventive practices, being between 25 and 59 years of age was associated with low preventive practices (compared with being less than 25 years of age) (AOR = 0.15, 95% CI 0.03–0.72). In contrast, achieving primary education (compared with a migrant worker with no formal education) (AOR = 3.24, 95% CI 1.18–8.93), and frequent acquisition of health information from all sources (AOR = 4.20, 95% CI 1.95–9.03) were significantly associated with high levels of preventive practices (Table 6). 

## 4. Discussion

Overall, we found that participants had high levels of awareness toward public health measures and preventive practices. The most common source of health information was social media. Based on the multivariate analysis, we found that a longer stay in Thailand had a significant association with lower awareness of public health measures. Frequent reception of health information and primary school completion showed a statistically significant association with a high level of preventive practices, relative to those who rarely received health information and had no formal schooling. However, being middle-aged (compared with being young) was significantly associated with fewer preventive practices. 

Our study also found that a longer stay in Thailand was associated with lower awareness about public health measures. One possible explanation might be that migrants who have lived in the host country for a longer period were more familiar with the lifestyle and may be less sensitive to new information. In addition, a longer stay was associated with multiple morbidities. To date, there is no clear explanation about this phenomenon but some studies in the literature have ascribed this to chronic stress, discrimination, or other factors associated with post-migration experiences [31,32].

A high frequency of receiving health information likely amplified the perception of risk and led to higher levels of preventive practices. Frequent communication via various channels had both direct and indirect effects on preventive behaviors, especially during the pandemic [33]. Therefore, the government should not overly rely on the delivery of health messages through a single particular channel but should distribute the information through multiple channels to ensure migrants engage in desired preventive behaviors.

From the findings above, middle-aged participants were likely to have lower levels of preventive practices compared with younger participants. According to the association between longer stay and low level of public health measure awareness, it could be inferred that as older migrants were more likely to have longer stay in the host country, they probably ignored or did not follow the public health measures. Evidence from other countries showed that migrants with a longer stay or those of older age tended to face greater health risks, compared with younger migrants [34]. Migrant workers with primary education were likely to exhibit higher degrees of preventive practices, compared with those with no formal education; having no formal education leads to low comprehension or misunderstandings about health messages, which may ultimately result in low levels of preventive practices [35]. Therefore, risk communication should not overlook migrants who have difficulties in accessing information due to low literacy, as well as those who are most likely to ignore or not conform to public health measures such as older migrants and migrants who have stayed in Thailand for more than 8 years.

This study was among the first studies to explore the importance of health risk communication for migrants during the COVID-19 pandemic. The use of empirical data was one of the main strengths of the study. However, it also contained certain limitations. First, we chose to use the median as a cut-off value to transform a multilevel categorical variable into a binary variable. Although this approach was able to facilitate the interpretation of results to a wide range of audiences, it might have also caused residual confounding. This issue also flags the need for further research, as there are no standard questionnaires to assess the level of health information reception among migrants thus far. Second, generalization of the findings is quite limited as the migrants in this study were mostly from Myanmar, while migrants in other settings (for example, those in the northeast or north of Thailand) are likely to have their own demographic characteristics and behaviors. Third, we could not guarantee our samples were free from selection bias. Despite our intent to ask local coordinators to randomly select migrants in the community to take part in the study, it is possible that the coordinators may have selected participants who were closer to them.

## 5. Conclusions

Among migrant workers in Thailand, the frequent reception of health information through various channels is positively linked with better preventive practices against COVID-19. Further initiatives to promote the education and health literacy of migrants should focus more on migrants with no formal education and migrants who are likely to ignore or not comply with public health measures. The government should deliver health messages through various channels to encourage migrants to engage in desired preventive behaviors. Further qualitative studies may overcome the limitations of this study.

## Figures and Tables

**Figure 1 ijerph-18-11474-f001:**
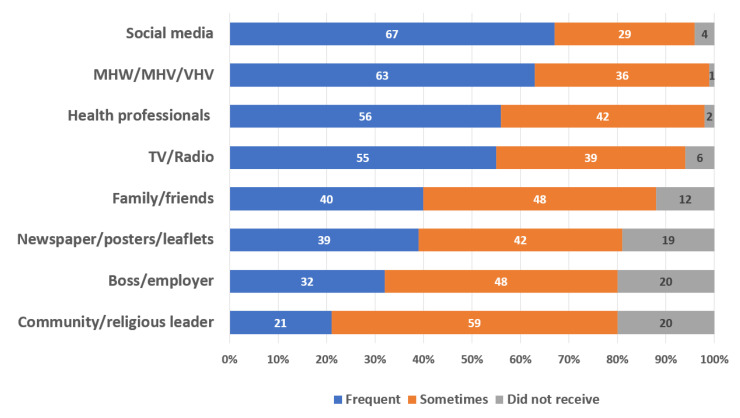
Proportion and frequency of health information received based on communication channels: MHW, migrant health worker; MHV, migrant health volunteer; VHV, village health volunteer.

**Figure 2 ijerph-18-11474-f002:**
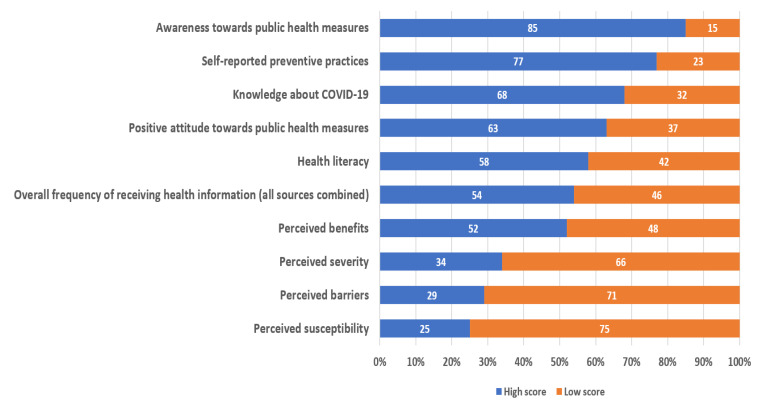
Proportion of scores for the variables.

**Table 1 ijerph-18-11474-t001:** Dependent variables and independent variables for analytic study.

Dependent Variables	Independent Variables
Awareness of public health measures on COVID-19	● Demographic data ● Source of information ○ From health professionals ○ From community members ○ From mass media ○ From social media
Self-reported preventive practices during COVID-19	Demographic data Resources for supporting preventive practices Health literacy Overall frequency of receiving health information from all sources Knowledge about disease Awareness and attitudes toward public health measures The four constructs of the health belief model (perceived risk, perceived benefit, severity of illness, and perceived barriers)

**Table 2 ijerph-18-11474-t002:** Characteristics of the variables and variable management.

Theme	Variables	Type of Variable	Classification
Demographic data	Gender	Categorical	Male Female
	Age (years)	Continuous, then changed to Categorical	≤24 25–59 ≥60
	Ethnicity	Categorical	Myanmar Non-Myanmar (Karen, Mon, Dawei, etc.)
	Length of stay in Thailand (years)	Continuous, then changed to categorical (cut point by median)	≤8 >8
	Education	Categorical	No formal education Primary school Secondary school and above
	Occupation	Categorical	Unemployed Factory worker Agriculture/fishery Construction worker Business/housemaid/waiter/other
	Family members (aged 15 years or above)	Continuous then changed to categorical (using the median as the cut-off point)	<2 people ≥2 people
	Income (minimum wage: Phuket 336, Ranong 315, Samut Sakhon 331 Baht/day ^ᵝ^)	Categorical then changed to categorical	Lower than minimum wage Equal to minimum wage Higher than minimum wage
	Thai Reading Comprehension Thai Listening Comprehension	Categorical	Fully understand Partially understand Cannot understand
	Health insurance	Categorical	Social Security Scheme ^a^ Health Insurance Card Scheme ^b^ No insurance/unknown
Access to resources	Electricity Phone signal Internet signal Face mask Soap Alcohol gel Tap water	Categorical	Good access Partial access Cannot access
	Resources supporting access to information (Total score for access to electricity, phone signal, and internet signal: 1–3 for each)	Categorical (using the median as the cut-off point)	Low access High access
	Resources supporting for preventive practices (Total score for access to face mask, soap, alcohol gel, and tap water: 1–3 for each)	Categorical (using the median as the cut-off point)	Low access High access
Health literacy	Health literacy (Total score for health literacy questions: 1–3 for each question, 12 questions in total) (Example: I can find information on health problems that concern me. (Agree/Neutral/Disagree))	Categorical (cut-off point of 60% of total score ᵞ)	Low health literacy High health literacy
Source of information	Sources Health professional (Thai staff) MHW/MHV/VHV Family/friends Community/religious leader Boss/employer Television/radio Newspaper/posters/leaflets Social media (Facebook, YouTube, etc.)	Categorical	Did not receive (score 1) Sometimes (score 2) Frequently (score 3)
	Score of health information sorted by source From health personnel (Total score for health professional and/or MHW/MHV/VHV: 1–3 for each) From community members (Total score for family/friends, community/religious leader, and boss/employer: 1–3 for each) From public mass media (Total score for television and radio, newspaper, posters, and leaflets: 1–3 for each) From social media (score of 1–3)	Categorical (using the median as the cut-off point)	Infrequent Frequent
	Overall score for frequency of receiving health information from all sources (Total score for all sources: 1–3 for each question, 8 questions in total)	Categorical (using the median as the cut-off point)	Infrequent Frequent
Knowledge	Knowledge about disease (Total score: 0–1 for each question, 12 questions in total) (Example: Loss of smell or taste is a symptom of COVID-19. (Yes/No))	Categorical (using the median as the cut-off point)	Low level of knowledge High level of knowledge
Awareness of public health measures	Awareness of public health measures (Total score: 0–1 for each question, 5 questions in total) (Example: If people visit Thailand from other countries, they must quarantine for at least 14 days. (Aware/Unaware))	Categorical (using the median as the cut-off point)	Low level of awareness High level of awareness
Attitude toward public health measures	Attitude toward public health measures (Total score: 1–3 for each question, 5 questions in total) (Example: Public gatherings are prohibited during COVID-19 outbreaks. (Agree/Neutral/disagree)	Categorical (using the median as the cut-off point)	Low level of positive attitude High level of positive attitude
Self-reported preventive practices	Regular preventive practices for COVID-19 situation (Total score: 0–1 of each, 7 questions in total) (Example: I always wear a mask when I go outside. (Yes/No))	Categorical (using the median as the cut-off point)	Low level of preventive practices High level of preventive practices
Health belief model	Perceived susceptibility (Total score: 1–3 for each question, 3 questions in total) (Example of questions: I consider myself to be at risk of COVID-19. (Agree/Neutral/Disagree))	Categorical (cut-off point at the 75th percentile ᵟ)	Low/moderate High
	Perceived severity (Total score: 1–3 for each question, 3 questions in total) (Example: If I get COVID-19, I will probably die. (Agree/Neutral/Disagree))	Categorical (cut-off point at the 75th percentile ᵟ)	Low/moderate High
	Perceived benefits (Total score: 1–3 for each question, 3 questions in total) (Example: I think that public health measures are good for me and my family. (Agree/Neutral/Disagree))	Categorical (cut-off point at the 75th percentile ᵟ)	Low/moderate High
	Perceived barriers (Total score: 1–3 for each question, 3 questions in total) (Example: I think that public health measures are problematic for my work. (Agree/Neutral/Disagree))	Categorical (cut-off point at the 75th percentile ᵟ)	Low/moderate High

^a^ A mandatory scheme financed by payroll taxes where employers, employees, and the government contribute equally. Migrants who have work permits are fully covered by this scheme. ^b^ A health insurance scheme for undocumented migrants managed by the Ministry of Public Health, Thailand. Migrants have to pay for the annual premium. ^β^ Source: Ministry of Labor, Thailand, data on 1 January 2020 [28]; ᵞ adapted from Simpson et al. [29]; ^δ^ adapted from Didarloo et al. [30].

**Table 3 ijerph-18-11474-t003:** Univariate analysis on awareness of public health measures on COVID-19.

Independent Factors	High Level of Awareness, n (%)	Low Level of Awareness, n (%)	*p*-Value
Gender			0.59
Female	157 (83)	33 (17)	
Male	83 (89)	10 (11)	
Age (years)			0.42
<25	31 (82)	7 (18)	
25–59	207 (85)	36 (15)	
≥60	3 (100)	0	
Ethnicity			0.72
Myanmar	155 (87)	24 (13)	
Non-Myanmar	84 (82)	19 (18)	
Province			0.59
Phuket	4 (80)	1 (20)	
Ranong	68 (91)	7 (9)	
Samut Sakhon	169 (83)	35 (17)	
Length of stay in Thailand (years)			0.03
≤8	135 (91)	13 (9)	
>8	103 (77)	30 (23)	
Education			0.44
No formal education	29 (88)	4 (12)	
Primary school	135 (82)	29 (18)	
Secondary school and upper	77 (89)	10 (11)	
Occupation			0.27
Unemployed	41 (77)	12 (23)	
Factory worker	119 (84)	22 (16)	
Agriculture/fishery	25 (93)	2 (7)	
Construction worker	38 (95)	2 (5)	
Business/housemaid/waiter	18 (78)	5 (22)	
Thai Reading Comprehension			0.15
Cannot understand	143 (81)	33 (19)	
Partially understand	84 (89)	10 (11)	
Fully understand	13 (100)	0	
Thai Listening Comprehension			0.41
Cannot understand	46 (90)	5 (10)	
Partially understand	134 (87)	20 (13)	
Fully understand	60 (77)	18 (23)	
Income			0.26
Lower than minimum wage	153 (83)	32 (17)	
Equal to minimum wage	13 (100)	0	
Higher than minimum wage	75 (87)	11 (13)	
Health insurance			0.10
No insurance/unknown	48 (76)	15 (24)	
Health Insurance Card Scheme	30 (83)	6 (17)	
Social Security Scheme	163 (88)	22 (12)	
Resources supporting access to information			0.96
Low access	23 (85)	4 (15)	
High access	217 (85)	38 (15)	
Family member (aged ≥ 15 years)			N/A
<2 people	14 (100)	0	
≥2 people	168 (82)	38 (18)	
Receiving health information from health professionals			0.37
Infrequent	117 (87)	18 (13)	
Frequent	124 (83)	25 (17)	
Receiving health information from community members			0.62
Infrequent	54 (84)	10 (16)	
Frequent	186 (85)	33 (15)	
Receiving health information from public mass media			0.75
Infrequent	20 (80)	5 (20)	
Frequent	220 (86)	37 (14)	
Receiving health information from social media			0.42
Infrequent	76 (80)	19 (20)	
Frequent	165 (87)	24 (13)	

**Table 4 ijerph-18-11474-t004:** Univariate analysis on self-reported preventive practices during COVID-19.

Independent Factors	High Level of Preventive Practices, n (%)	Low Level of Preventive Practices, n (%)	*p*-Value
Gender			0.04
Female	140 (73)	51 (27)	
Male	79 (85)	14 (15)	
Age (years)			0.03
<25	35 (95)	2 (5)	
25–59	182 (74)	63 (26)	
≥60	3 (100)	0	
Ethnicity			0.09
Myanmar	145 (81)	34 (19)	
Non-Myanmar	74 (70)	31 (30)	
Province			0.45
Phuket	8 (100)	0	
Ranong	56 (77)	17 (23)	
Samut Sakhon	156 (76)	48 (24)	
Length of stay in Thailand (years)			0.90
≤8	117 (78)	33 (22)	
>8	101 (77)	31 (23)	
Education			0.05
No formal education	21 (64)	12 (36)	
Primary school	128 (77)	38 (23)	
Secondary school and upper	71 (83)	15 (17)	
Occupation			0.03
Unemployed	34 (65)	18 (35)	
Factory worker	111 (79)	30 (21)	
Agriculture/fishery	18 (64)	10 (36)	
Construction worker	38 (93)	3 (7)	
Business/housemaid/waiter	19 (83)	4 (17)	
Thai Reading Comprehension			<0.01
Cannot understand	126 (71)	51 (29)	
Partially understand	80 (85)	14 (15)	
Fully understand	13 (100)	0	
Thai Listening Comprehension			0.65
Cannot understand	38 (73)	14 (27)	
Partially understand	123 (80)	31 (20)	
Fully understand	58 (74)	20 (26)	
Income			0.16
Lower than minimum wage	142 (76)	44 (24)	
Equal to minimum wage	5 (38)	8 (62)	
Higher than minimum wage	73 (85)	13 (15)	
Health insurance			<0.01
No insurance/unknown	39 (62)	24 (38)	
Health Insurance Card Scheme	30 (83)	6 (17)	
Social Security Scheme	151 (81)	35 (19)	
Resources supporting access to preventive practices			0.39
Low access	37 (77)	11 (23)	
High access	181 (77)	54 (23)	
Family member (age ≥ 15 years old)			0.31
<2 people	12 (86)	2 (14)	
≥2 people	149 (72)	58 (28)	
Overall frequency of receiving health information (all sources combined)			<0.01
Infrequent	88 (68)	42 (32)	
Frequent	129 (85)	22 (15)	
Health literacy			0.73
Low health literacy	91 (77)	27 (23)	
High health literacy	127 (77)	37 (23)	
Positive attitude toward public health measures			0.69
Low level of positive attitude	80 (78)	23 (22)	
High level of positive attitude	136 (77)	41 (23)	
Knowledge about disease			0.33
Low level of knowledge	65 (72)	25 (28)	
High level of knowledge	155 (79)	40 (21)	
Awareness of public health measures			0.33
Low level of awareness	28 (65)	15 (35)	
High level of awareness	188 (79)	50 (21)	
Perceived susceptibility			0.08
Low/moderate perception	160 (75)	52 (25)	
High perception	60 (83)	12 (17)	
Perceived severity			0.10
Low/moderate perception	140 (74)	49 (26)	
High perception	79 (83)	16 (17)	
Perceived benefits			0.83
Low/moderate perception	102 (76)	33 (24)	
High perception	117 (79)	32 (21)	
Perceived barriers			0.03
Low/moderate perception	149 (74)	53 (26)	
High perception	70 (85)	12 (15)	

**Table 5 ijerph-18-11474-t005:** Multivariate analysis on the awareness of public health measures during COVID-19.

Selected Factors	Adjusted OR (95% CI)	*p*-Value
Length of stay in Thailand		
≤8 years	reference	
>8 years	0.43 (0.19–0.95)	0.04
Insurance		
No insurance/unknown	reference	reference
Social Security scheme	2.02 (0.81–5.06)	0.13
Health insurance card scheme	1.78 (0.54–5.85)	0.35

**Table 6 ijerph-18-11474-t006:** Multivariate analysis on self-reported preventive practices during COVID-19.

Selected Factors	Adjusted OR (95% CI)	*p*-Value
Gender		
Female	reference	
Male	1.08 (0.46–2.54)	0.86
Age (years)		
<25	reference	
25–59	0.15 (0.03–0.72)	0.02
≥60	NA	NA
Ethnicity		
Myanmar	reference	
Non-Myanmar	0.63 (0.29–1.35)	0.24
Education		
No formal education	reference	
Primary school	3.24 (1.18–8.93)	0.02
Secondary school and upper	2.64 (0.85–8.16)	0.09
Occupation		
Unemployed	reference	
Factory worker	0.46 (0.09–2.49)	0.37
Agriculture/fishery	0.38 (0.09–1.59)	0.18
Construction worker	2.20 (0.31–15.51)	0.43
Business/housemaid/waiter	0.77 (0.16–3.59)	0.74
Thai Reading Comprehension		
Cannot understand	reference	
Partially understand	1.71 (0.72–4.05)	0.22
Fully understand	NA	NA
Insurance		
No insurance/unknown	reference	
Health insurance card scheme	2.46 (0.64–9.43)	0.19
Social Security scheme	2.49 (0.58–10.62)	0.22
Overall frequency of receiving health information (all sources combined)		
Infrequent	reference	
Frequent	4.20 (1.95–9.03)	<0.01
Perceived susceptibility		
Low/moderate perception	reference	
High perception	0.98 (0.37–2.60)	0.97
Perceived severity		
Low/moderate perception	reference	
High perception	1.18 (0.47–3.00)	0.72
Perceived barriers		
Low/moderate perception	reference	
High perception	0.97 (0.38–2.48)	0.95

Note: NA = dropped from the analysis due to perfect prediction.

## Data Availability

The datasets used in this article are not available publicly; however, they may be available upon reasonably request to IHPP.

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
