# Peer review of "Factors Related to Health Risk Communication Outcomes among Migrant Workers in Thailand during COVID-19: A Case Study of Three Provinces"

_ijerph, 2021, doi:10.3390/ijerph182111474_

Round 1
Reviewer 1 Report
Abstract. OK
Introduction. Very clear and well justified. I have only one comment, risk communication was not a study variable. It was used to contextualize the study and it was correctly used in the title and the objective of the abstract, but not in the introduction. It says: “to explore the outcomes of health risk communication …” (line 74). It should say: to explore “factors related to the outcomes of health risk communication…” (same as in the abstract).
Material and methods. Could you please give an example of at least one question on health literacy, knowledge, awareness of public health measures, attitude, self-reported preventive practices; and perceived susceptibility, benefit, and barriers (Table 2).
Results. OK
Discussion. OK
Conclusions. Risk communication was not a study variable. It is not possible to conclude "The adequacy of information received and higher frequency of risk communication… were positively linked to better preventive practices…" (line 284), since such qualities were not measured and not used in the statistical models. Please adjust the writing accordingly.
Reviewer 2 Report
Abstract
No reference to covid-19 in the abstract. Hard to understand the work by just reading the abstract. Numerous errors in the abstract.
Line 17. This study used...... not appropriate instead it should be "we conducted a cross-sectional survey".......
Line 19. Bilingual questionnaire -not clear. Language use should be specified. It appears that the questionnaire is bilingual.
Line 19. should be rewritten- not clear
line 22. Sample size info should come before the results
Line 22. High exposure-----not clear. should be rewritten
line 24-28. not clear should be rewritten
The abstract should be completely overhauled.
Introduction
Was not presented in coherent fashion and repleted with errors. No enough background information provided. Hard to follow what the authors seek to accomplish. Error after error. Needs significant language edition. Justification for the study not clearly presented.
Method section
2.1. Study design. Actually it wasn't study design at all. Three different sample sizes mentioned.
Line 123-131 not clear. Significant language edition required
Data analysis. Not clear
Table 1. Means of receiving information: better to say: source of information
Table 2. Family members (aged since 15 years old and older): hard to understand
Results section.
Needs significant edition.
Discussion section: needs language edition
Conclusion looks okay.
Significant problem. Ethical approval of research involving human subjects missing.
Round 2
Reviewer 2 Report
Dear authors,
Attached here please find my comments.
